# The Oral Health of a Group of 19th Century South Australian Settlers in Relation to Their General Health and Compared with That of Contemporaneous Samples

**DOI:** 10.3390/dj11040099

**Published:** 2023-04-07

**Authors:** Angela Gurr, Maciej Henneberg, Jaliya Kumaratilake, Derek Lerche, Lindsay Richards, Alan Henry Brook

**Affiliations:** 1Discipline of Anatomy and Pathology, School of Biomedicine, University of Adelaide, Adelaide, SA 5005, Australia; 2Biological Anthropology and Comparative Anatomy Research Unit, School of Biomedicine, University of Adelaide, Adelaide, SA 5005, Australia; 3Institute of Evolutionary Medicine, University of Zurich, 8006 Zurich, Switzerland; 4School of Dentistry, University of Adelaide, Adelaide, SA 5005, Australia; 5Institute of Dentistry, Queen Mary, University of London, London WC1E 7HU, UK

**Keywords:** oral health, systemic health, colonial dental health

## Abstract

The aims of this study are to determine the oral health status of a rare sample of 19th-century migrant settlers to South Australia, how oral conditions may have influenced their general health, and how the oral health of this group compares with contemporaneous samples in Australia, New Zealand, and Britain. Dentitions of 18 adults and 22 subadults were investigated using non-destructive methods (micro-CT, macroscopic, radiographic). Extensive carious lesions were identified in seventeen adults and four subadults, and from this group one subadult and sixteen adults had antemortem tooth loss. Sixteen adults showed evidence of periodontal disease. Enamel hypoplastic (EH) defects were identified in fourteen adults and nine subadults. Many individuals with dental defects also had skeletal signs of comorbidities. South Australian individuals had the same percentage of carious lesions as the British sample (53%), more than other historic Australian samples, but less than a contemporary New Zealand sample. Over 50% of individuals from all the historic cemeteries had EH defects, suggesting systemic health insults during dental development were common during the 19th century. The overall oral health of the South Australian settlers was poor but, in some categories, (tooth wear, periapical abscess, periodontal disease), better than the other historic samples.

## 1. Introduction

A skeletal sample of 19th-century migrant settlers to South Australia was investigated to understand their health status. There are few collections of skeletal remains from early migrants to Australia. The majority of the individuals from this South Australian sample were buried in the first three decades after the establishment of the colony in 1836. Skeletal abnormalities in these individuals have been examined and the findings in relation to the general health of the individuals have been published [1,2]. The dentoalveolar complexes of these skeletons were investigated in this study to expand our understanding of the health status of the individuals in relation to their dental and oral health.

Tooth enamel cannot be remodeled in life [3,4]. Dentine can be remodeled but only very slowly as a result of the aging process and/or due to dental caries or trauma [3,4]. Therefore, health insults that occur during dental development in utero and postnatally until young adulthood [5,6] can cause enamel and dentine defects such as enamel hypoplasia and interglobular dentine that remain throughout life [4]. This makes dentition an excellent model for the investigation of an individual’s health history compared with bones, which become remodeled during life in response to changes in the forces acting on them [7]. Skeletal abnormalities that result from a health insult in early life may alter or change with the remodeling of the bone.

The presence of carious lesions, morphological changes to the alveolar bone, including its loss, and antemortem tooth loss due to extraction, disease, or trauma are all valuable sources of evidence. As a result of teeth being in direct contact with their environment, patterns of tooth wear from attrition, abrasion, and erosion can provide information regarding diet, lifestyle choices, oral hygiene, the use of teeth in daily activities, and the environment of the individual. Cultural practices affecting the oral environment may also leave their mark on dentition, for example, pipe mouthpieces or pins and nails habitually held between teeth by some tradespeople. The status of the oral health of this group of early settlers could also give an indication of the degree to which they had access to the available dental services [8].

The relationship between poor oral health and systemic ill health has been investigated [9,10,11]. Periodontal disease has links to many systemic illnesses, such as atherosclerotic cardiovascular disease [12,13,14,15], Type 2 diabetes [9,16], and bacterial pneumonia [17,18]. Considering the oral health of the St Mary’s Cemetery sample, together with any skeletal signs of disease to indicate comorbidities [1,2], should provide the clearest possible insight into the health of these individuals.

Therefore, the aims of this study were to investigate (1) the oral health status of the individuals buried at the ‘free ground’ of St Mary’s Anglican Church Cemetery, (2) how these oral conditions may have influenced their general health and, (3) how the oral health status of these colonial South Australian settlers compared with other historic samples in Australia, New Zealand, and Britain.

## 2. Materials and Methods

### 2.1. Materials—The Archaeological Sample

The 70 individuals in this sample were buried between 1847 and 1927, in an unmarked area of St Mary’s Cemetery, referred to as the ‘free ground’, which was for people who had no funds to pay for their interment. These burials were paid for by the South Australian Government. Burials in the main section of the cemetery were paid for by the deceased or their family; these burials came with a gravestone [1].

To identify the individuals excavated from the free ground, each skeleton was assigned a site code and number (e.g., St Mary’s Burial/number 58 = SMB 58). No other sections of the cemetery have been excavated. Relevant aspects concerning the excavation of this sample and the observed macroscopic skeletal abnormalities have been published [1,2,19,20]. Determination of the sex of each individual and an estimation of the age range at death, from the skeletal remains and macroscopic examination of tooth eruption, were recorded immediately after the excavation of each skeleton. [19].

#### 2.1.1. The Archaeological Sample—Dentitions

The skeletal remains of 40 individuals where dentitions were preserved in a suitable state were selected for the study. This sample consisted of 18 adults (13 male and 5 female) and 22 subadults from the St Mary’s Cemetery skeletal collection (total *n* = 70). The terms ‘infant’ and ‘subadult’ are used in this study to refer to individuals in the following age ranges: 0–2 years (infants); 3–5 years, 6–9 years, 10–15 years, and 16–19 years (all groups of subadults). Appendix A provides an example using these age categories.

The maxillary and mandibular bones of the remaining 30 individuals within this sample were highly fragmented and not suitable for analysis. The specimens investigated comprised dentitions in situ within the dentoalveolar complex of the skull and had lost teeth that had been displaced post-mortem. A dental inventory (number and type of teeth present), an estimation of the age range (assessment of the skeletal remains and dentitions), and the sex of each individual are presented in Appendix A.

#### 2.1.2. The Archaeological Sample—Ethics

The excavation of the free ground section of St Mary’s Anglican Church Cemetery occurred upon the request of the Parish as they wished to re-use the area. Approval for the study of the skeletal remains was also granted by St Mary’s Parish and the Flinders University Social and Behavioural Research Ethics Committee (SBREC project number 8169). Destructive analysis was not permitted during the investigation of this sample as the remains are of a rare historical nature.

### 2.2. Methods

Large Volume and Small Volume (LV or SV Micro-CT) scanning systems, macroscopic examination, and standard dental radiographs were used for the investigation of the structures of the dentoalveolar complexes of the selected human skulls from the St Mary’s skeletal collection.

#### 2.2.1. Large Volume Micro-CT

This technique allowed for the examination of ‘large ‘specimens such as dentitions that remain in situ within the skull that require minimal handling. This method provides rapid data acquisition at a high resolution of both the external and internal structures [21]. The dentoalveolar complexes in situ within the skulls of six individuals (two adults, SMB 66B and SMB 73; two subadults, SMB 04A and SMB 52B; and two infants SMB 58 and SMB 82) were scanned using the Nikon XT H 225 ST cabinet Micro-CT scanning system [22]. The pixel/voxel size (spatial resolution) used for each scan was different as these are relative to the size of the specimen. The settings used for each complete skull, cranium, or mandible alone are presented, with additional scanning information, in Appendix A.

#### 2.2.2. Small Volume Micro-CT

Teeth that had been displaced post-mortem from 21 individuals (*n* = 41 teeth) (Appendix A) were investigated. The number of teeth and tooth types that were available for each individual varied. Individual tooth specimens were scanned using the desktop Bruker SkyScan 1276 [23], at a pixel size of 9.0 µm. Information regarding the SV Micro-CT scan settings is presented in Appendix A.

#### 2.2.3. Macroscopic Examination

Visual investigation of the structures of the dentoalveolar complexes was conducted in a dry laboratory with the aid of a table magnifying glass with enhanced lighting. A periodontal probe with incremental markings was used for measurements [24]. The Fédération Dentaire Internationale (FDI) World Dental Federation notation system [25,26] was used in this study to accurately record and distinguish between permanent and primary teeth.

#### 2.2.4. Standard Dental Radiographs

Panoramic extraoral radiographs (crania and mandibles were imaged separately) and periapical intraoral radiographs were taken using Planmeca X-ray equipment [27]. Details of the radiographic equipment and the settings used are available in the Appendix A in Appendix B. Dental radiographs can only identify severe cases of interglobular dentine (IGD) or enamel hypoplastic (EH) defects, therefore these categories were not scored on those.

#### 2.2.5. Scoring: Dental and Alveolar Bone Health Categories

The following dental and alveolar health categories were scored: dental inventory, dental age, tooth wear, presence of carious lesions, periodontal disease, enamel hypoplastic (EH) defects, and interglobular dentine (IGD). The identification criteria and scoring systems used for the above categories are listed in Table 1. Teeth with more wear could be suggestive of an older adult. Therefore, 11 adults from the St Mary’s sample with permanent molars remaining in situ were scored using Miles’s (1962) [28] tooth wear system for archaeological specimens. This system assesses the functional age of each molar and predicts the age of the subject [28]. To determine the extent of general tooth wear, Molnar’s (1971) [29] system was also used for all individuals in the sample.

Avizo 9 data visualisation software [30] was used for image analysis of the reconstructed LV and SV Micro-CT scan data sets. The software (Avizo 9) provides digitally reconstructed radiographs (DRRs) from the LV and SV Micro-CT scans which were compared to the dental radiographs. The category of radiolucency of caries (Table 1) was scored on both the dental radiographs and the DRRs. This software was used to manipulate the bone density threshold levels of the specimens (i.e., reduce the density of the alveolar bone) to reveal developing teeth.

Alveolar bone loss, which was suggestive of periodontal disease, was scored using a measurement from the cementoenamel junction (CEJ) on a tooth to the crest of the alveolar bone [24]. This measurement was taken along the midline of the tooth on the labial/buccal and lingual/palatal surfaces. An increase in the distance measurement from the CEJ to the crest of the alveolar bone suggested that bone loss had occurred. Measurements of 4 mm or above are included in this analysis.

The St Mary’s sample were examined for evidence of calculus deposits. Adult SMB 84 is edentulous (with dentures) and not included in this analysis. The presence and location of the calculus on the tooth (enamel or root) and the severity of the deposits (small/slight, medium, or large/considerable) were scored following the criteria set out by Connell and Rauxloh (2003) [31] and Powers (2012) [32] from the Museum of London Human Osteology Method Statement. These systems are developed from that of Brothwell (1963) [33].

It is difficult to differentiate areas of staining on the tooth that could be caused by the burial environment and/or taphonomic changes from developmental enamel opacities and/or variations in the mineralisation of the enamel [34,35,36]. Therefore, enamel opacities (hypo-mineralisation defects) were not investigated.

**Table 1 dentistry-11-00099-t001:** Categories for scoring dental and alveolar bone health.

Categories to Be Investigated	Criteria for Identification	Scoring Systems	References
**Dental** **Inventory**	Total number of teeth in situ. Antemortem tooth loss: evidence of alveolar tissue healing. Postmortem tooth loss: open socket and no evidence of bone healing	Data were recorded on a visual chart representing the primary/permanent teeth using the FDI (ISO 3950) notation system. (i) tooth type present, (ii) location of healed alveoli, (iii) open socket—location in the alveolar process	[25,26,37,38,39,40]
**Dental** ** age range**	Erupted tooth types present, semi-erupted, and developing teeth in alveolar bones Tooth wear—adult molars only	The London Atlas of tooth eruption and development was used with dental radiographs to identify the stage of eruption and tooth development (0–23.5 years). Adult age range: assessment of the functional age of each molar and the predicted age of the subject based on tooth wear scores set out by Miles (1962).	[5,6,28]
**Tooth wear**	Evidence of enamel loss and/or exposure of dentine on the occlusal surface of the teeth	Category of tooth wear selected from Molnar’s (1971) and Miles’s (1962) criteria charts.	[28,29]
**Carious lesions** (**caries—cavity**)	(1) Evidence of decay: (a) present on enamel surface only, (b) involving enamel and dentine, (c) decay involving the enamel dentine and the pulp. (2) Identify changes in radiolucency/density of the tooth	Score: (i) tooth type affected (FDI), (ii) location of the carious lesion in relation to the CEJ, (iii) ICDAS/ICCMS category of radiolucency-using dental radiographs and DRRs. Select a category from a visual chart.	[41]
**Periodontal disease**	(i) Evidence of alveolar bone loss (ii) Evidence of morphological changes in the margins of the contours of the alveolar bone of the posterior teeth (buccal surface only)	(i) Measurement taken from the CEJ to the crest of the alveolar bone on the midline of the crown surface (labial/buccal and lingual/palatal). (ii) Alveolar bone status: graded 0–4 using Ogden’s (2008) system via inspection of the margins of the alveolar bone surrounding the posterior teeth.	[42,43,44]
**Enamel hypoplastic defects** (**EH**)	Evidence of lines or pits in the surfaces of the enamel	Scored using an adaptation of the Enamel Defect Index (EDI): (i) type of EH defect/s, (ii) number of EH defects on the enamel surface, (iii) location of EH defect/s—measurement of the distance of the defect/s in relation to the CEJ.	[45,46]
**Interglobular dentine** (**IGD**)	Evidence of changes in the density of the dentine structure	Record: the presence of IGD as Yes/No (Micro-CT only)	[47,48]

Notes: CEJ = Cemento-enamel Junction, FDI = Fédération Dentaire Internationale/World Dental Federation notation system, DRR = Digitally Reconstructed Radiographs using Avizo 9 software [30], ICDAS/ICCMS = International Caries Detection and Assessment System/International Caries Classification and Management System.

#### 2.2.6. Scoring: Evaluation of Intra and Inter-Operator Variations

##### Intra-Operator Variation

The primary operator (AG) macroscopically examined and scored the images of teeth and alveolar tissues using dental radiographs and 2D and 3D images taken from the LV and SV Micro-CT scans using the Avizo 9 software [30]. A second data scoring session was carried out using the same specimens two weeks later for the evaluation of the intra-operator variability.

##### Inter-Operator Variation

Operators were trained and calibration sessions were carried out two weeks before these inter-operators independently scored the dental and alveolar bone health categories (Appendix A).

#### 2.2.7. Statistical Analysis

All analyses were performed using Stata v17 computer software [49]. Assessments of the intra-operator and inter-operator reliability were made using Gwet’s Agreement Coefficient (AC1), weighted Gwet’s Agreement Coefficient (AC2), and Intraclass Correlation Coefficient (ICC) using a two-way random-effects model for absolute agreement, for binary and nominal scale data, and ordinal scale data and continuous data, respectively. Results are presented as AC1/AC2 with a 95% confidence interval (CI) and percentage agreement for non-continuous data and as the ICC with a 95% CI for continuous data. Interpretation of AC1 and AC2 was <0 = poor agreement; 0–0.2 = slight agreement; 0.21–0.4 = fair agreement; 0.41–0.6 = moderate agreement; 0.61–0.8 = substantial agreement; and >0.8 = almost perfect agreement [50]. Interpretation of ICC values was <0.50 = poor agreement; 0.50–0.75 = moderate agreement; 0.75–0.90 = good agreement; and >0.90 = excellent agreement [51].

### 2.3. Comparison of Historic Dental Samples from Australian, New Zealand, and British Cemeteries

Findings from the investigation of the dental and alveolar bone health categories for the St Mary’s Cemetery sample were compared with data from two colonial Australian samples: Cadia Cemetery, NSW, 1864–1927 (*n* = 109) [52] and Old Sydney Burial Ground (OSBG), NSW, 1792–1820 (*n* = 10) [53]. Dental findings from Cadia Cemetery have not been previously published; therefore, permission was granted by the copyright holder, Newcrest Mining Ltd. and Dr. Edward Higginbotham and Associates Pty Ltd., the consultant archaeologist. The St Mary’s findings were also compared with the published data from St John’s Burial Ground, Milton, Otago, New Zealand, (1860–1926) (*n* = 7) [54] and a British sample from the Cross Bones Burial Ground, Southwark, London, UK (1800–1853) (*n* = 83) [55]. The category of tooth wear for each cemetery was scored using different systems; therefore, only scores that represented ‘moderate to heavy’ tooth wear were compared.

## 3. Results

### 3.1. Reproducibility—Standard Statistical Analysis

Due to the small sample size, many tests for intra-operator and inter-operator agreement achieved perfect agreement, resulting in all summary ranges, including perfect or excellent agreement [50,51]. For the inventory and tooth wear measurements, the inter-operator agreement was better when using the Macroscopic or Radiographic techniques; the results ranged from poor to perfect for the Macroscopic technique and substantial to perfect for the Radiographic technique, compared with fair to perfect for the LV Micro-CT technique; for both Macroscopic and Radiographic techniques combined, the results ranged from moderate to perfect, compared to fair to perfect for the LV Micro-CT technique (for inventory and tooth wear, respectively). For alveolar status, the inter-operator agreement was the same for both the LV Micro-CT and Macroscopic techniques. The LV Micro-CT method achieved the same level of agreement as the other techniques for all other measures when assessing both the inter- and intra-operator agreement. The Macroscopic method also had an almost perfect agreement. It should be noted though that the percentage agreement was very similar—between 81 and 100% agreement for the LV Micro-CT method and 88 to 100% agreement for the Macroscopic method.

A summary of the tests of intra- and inter-operator reliability for each of the methods investigated is presented in Appendix A. A written summary with information on this statistical analysis follows Appendix A. Additional data and the raw data can be found in Appendix A, respectively.

### 3.2. Dental Inventory

A full dental inventory for each individual is presented in Appendix A. Thirty-nine of the forty individuals had dentitions in situ. The total number of teeth present for the sample (*n* = 40) was 518 (175 primary and 343 permanent teeth). There were eight adults, and each had less than ten teeth present (Appendix A). One adult female, SMB 84, was edentulous and had a full set of vulcanite dentures with porcelain teeth (Figure 1). The setup/fabrication of the dentures suggests that they were well made.

#### Dental Age Range

The estimated dental age range for each individual is presented in Appendix A. The dental age range of the majority of individuals was similar to that assigned from the skeletal assessment (Anson, 2004). The dental age ranges of the two subadults (SMB 28, SMB 70) were different from the skeletal age range (Appendix A). Subadult SMB 28: the dental age range was 15.5–16.5 years (±1 year) compared with the 12–13 years skeletal age range. The skeletal age was based on the incomplete fusion of the epiphyseal plates at the elbow [19]. Radiographically, there was no evidence of the mandibular third molars developing, and the alveolar tissue in this area was fragmented. The developing crowns of these molars were loose and had separated from the jaw. Subadult SMB 70: the dental age range was 11.5–12.5 years (±1 year), compared with the 8–9 years skeletal age range.

### 3.3. Tooth Wear

In scoring tooth wear, a higher category implies more wear [28,29]. The functional age of the permanent molars of 11 adults was scored to estimate the age range of each individual [28] ( S6). Some of the age ranges predicted using Miles’s (1962) system [28] were different from the age ranges assigned to the individuals based on the evidence of skeletal changes/maturity and dental eruption. Molnar’s (1971) [29] system provided general tooth wear scores for each tooth present in all of the individuals in the sample (Appendix A). The individuals with higher categories of tooth wear (categories 4–8) [29] were all adults (Appendix A). Analysis of the distribution of tooth wear for the different tooth types (i.e., incisors, canines, premolars, etc.) showed that more anterior teeth (central and lateral incisors as well as canines) were scored with category 4 and 5 tooth wear compared with premolars and molars (Table 2). The canine teeth of the St Mary’s adults were the only anterior teeth to be scored with category 5 tooth wear, along with the molars (Table 2).

Three adult males (SMB 59, SMB 72, and SMB 78) showed patterns of tooth wear that suggest they had smoked a pipe for an extended period of time. The opposing upper and lower teeth, for example the maxillary and mandibular canines and first premolars, had a semi-circular pattern of tooth wear, suggesting these teeth could have gripped a pipe (Figure 2).

### 3.4. Carious Lesions

Seventeen adults and four subadults (Table 3) had evidence of carious lesions on their dentitions (*n* = 21/40–53%). Seven individuals from this group had more than 60% of their teeth affected by caries (Table 3). The majority of the carious lesions were seen on the permanent teeth, except for the primary teeth of subadults SMB 19 (FDI tooth numbers 55 and 83) and SMB 70 (FDI tooth numbers 53 and 63). The total number and percentage of teeth affected by carious lesions are presented in Table 3. A greater number of lesions were located *on* the CEJ (46%) rather than above (42%) or below the CEJ (12%) (Appendix A), and a greater number of carious lesions was located on the mesial (29%) and distal (25%) surfaces of the teeth than on the occlusal (14%), the labial/buccal (21%), and the lingual/palatal (11%) surfaces (Appendix A). The extent of the carious lesions (e.g., how many involved the enamel only, the enamel and the dentine, or were extensive and approaching the pulp chamber) is presented in (Appendix A). The majority of carious lesions observed in the St Mary’s sample involved the enamel only and had not extended to involve the dentine or pulp (Appendix A).

A large carious lesion was identified on the mesial/occlusal surface of the lower right permanent first molar (M1) of subadult SMB 79 (Figure 3a). Resorption of the alveolar bone surrounding this tooth exposed the majority of the buccal roots (Figure 3b). A localised periapical cavity was present with a circular hole (fistula) in the alveolar bone (3 mm in diameter) on the buccal surface, inferior to the mesial root of the M1 (Figure 3b,c). Evidence of changes in the texture of the alveolar bone surface (porosity) around this fistula was seen (Figure 3b). Radiographs showed the carious lesion had involved the enamel, dentine, and pulp of this tooth. The X-ray image also shows the extent and depth of the periapical cavity around the distal root of the M1 and the ‘opening’ in the alveolar bone adjacent to the apex of the mesial root (Figure 3c).

### 3.5. Periodontal Disease

#### 3.5.1. Alveolar Bone Loss

Nine out of forty adults (seven males: SMB 06, SMB 23, SMB 57, SMB 63, SMB 68, SMB 73, SMB 78, and two females: SMB 53C, SMB 66B) aged between 30–49 years old had a distance measurement from the CEJ to the crest of the alveolar bone of 4 mm or over (up to 10 mm) on three or more teeth, suggesting evidence of periodontal disease. Adult SMB 73 had 9 teeth with distance measurements ranging from 6 mm to 10 mm (other affected teeth measured from 4 to 5 mm). Two adults (SMB 14 and SMB 85, both male, aged between 40–59 years old) only had two teeth remaining having lost the majority of their teeth in life (antemortem). These teeth showed extensive horizontal alveolar bone loss (between 6–7mm per tooth). The female with the dentures (SMB 84) (Figure 1) had complete resorption of alveolar processes.

#### 3.5.2. Alveolar Bone Status:

Ogden’s (2008) [42] system of assessment is as follows: grade 0 = unable to score, i.e., alveolar is missing, grade 1 = no disease and grade 4 = severe periodontitis. The higher grades (3 and 4) suggestive of evidence of periodontal disease are shown here.

**Grade 3**—*n* = three adults (males, SMB 72, SMB 73, and SMB 83), aged between 30–59 years old.

**Grade 4**—*n* = eight adults (seven males, SMB 06, SMB 09, SMB 23, SMB 57, SMB 63, SMB 68, and SMB 85, and one female, SMB 66B), aged between 30–39 years old.

The buccal contours of the alveolar margins of the posterior teeth for adult SMB 06 (aged between 40–49 years old) had the highest number of areas scored at grade 4. The remaining areas of alveolar bone for this individual were scored from grade 1 to 3, suggesting extensive periodontal disease.

#### 3.5.3. Calculus

Calculus (calcified plaque) deposits were observed on the remaining dentitions of 11 of the 17 adults from the St Mary’s sample; the full results per tooth are presented in Appendix A. A total of 240 teeth were examined, with 79 teeth (33%) being affected by calculus (Table 4 and Appendix A). The calculus deposits observed were of small (slight)-to-medium amounts (Appendix A). Five adults from the group of eleven had between 68% and 100% of their teeth affected by calculus (Appendix A). The location of the calculus was predominantly on the enamel surface (60/79 teeth) (Appendix A).

### 3.6. Enamel Hypoplasia (EH)

Fourteen adults and ten subadults showed evidence of one or more enamel hypoplastic defects on one or more of their teeth (Table 5a and Appendix A), representing over half of the total sample (*n* = 24/40–60%). Four adults and four subadults had EH defects on 60% or more of their teeth (Table 5). Adult SMB 73 had the highest percentage of teeth affected by EH defects (Table 5). More canine teeth were affected by EH defects than the incisors, premolars, or molars for both primary and permanent dentition (Table 6 and Appendix A); this was followed by the central and lateral incisors. The permanent third molars were the least affected tooth type (Table 6). Large Volume Micro-CT scans identified EH defects on the erupted primary teeth and the developing permanent teeth of infant SMB 58 (Figure 4) (Table 5). This infant also had evidence of EH defects on four primary canines, four primary second molars, and sixteen permanent teeth (central and lateral incisors, canines, and first molars), either starting to erupt or developing within the alveolae of the maxilla and mandible (Figure 4) (Table 5 and Appendix A). Details of the number of EH defects per tooth per individual are presented in Appendix A.

### 3.7. Interglobular Dentine (IGD)

Two individuals (adult SMB 73 and infant SMB 58) of the six individuals that were scanned using the LV Micro-CT system showed areas of IGD. This mineralisation defect in the dentine was seen in all of the erupted and developing primary teeth of infant SMB 58 and of the permanent dentition within the alveolae (developing maxillary and mandibular teeth listed above) (Figure 5). Individual SMB 73 had IGD in 17 of his 19 permanent teeth. Three individuals (adults SMB 63 and 73 and subadult SMB 70) from the twenty-one individuals who had one or more loose teeth scanned using the Small Volume Micro-CT system (Appendix A) showed IGD.

### 3.8. Comorbidities and Signs of Skeletal and Dental Changes

Twenty-five individuals from the St Mary’s dental sample of forty had poor oral health. Their oral health findings are presented in Table 7. From this group, one infant, five subadults, and seven adults (*n* = 13/25) also had skeletal signs of one or more comorbidities and/or signs of skeletal abnormalities and/or evidence of dental changes, including indicators of lifestyle habits (Table 7). Some comorbidities are only evident in the soft tissue [9,12,13,14,15,16] and could not be identified in the skeletons of this sample. Seven adults had evidence of skeletal abnormalities such as Schmorl’s nodes and vertebral osteophytes (Table 7) [1,2] and three adult males had dental changes due to tooth wear (pipe notch), which indicate smoking and an increased risk of poor oral health.

### 3.9. Comparison of Historic Dental Samples from Australian, New Zealand, and British Cemeteries

#### 3.9.1. Demographic Profiles

The Australian, New Zealand, and British samples, compared with the St Mary’s Cemetery sample, are presented in Table 8 and Table 9. There were differences and similarities between the samples. For example, (a) no subadults were included in the dental sample from the Old Sydney Burial Ground (OSBG), NSW, or the St John’s Burial Ground, NZ, (Table 8); (b) the percentage of subadults in the cemeteries of St Mary’s, SA and Cadia, NSW, Cross Bones, UK was similar. The St Mary’s and Cadia cemeteries had a slightly higher percentage of subadults (55% and 67%, respectively) than adults compared with the Cross Bones Burial Ground (Table 10).

#### 3.9.2. Dental and Alveolar Bone Health Categories

A summary of the scores for the five oral health categories of St Mary’s sample were compared with those of the Cadia Cemetery, Old Sydney Burial Ground (OSBG), St John’s, and Cross Bones burial grounds (Table 10). All of the individuals from St John’s were scored with ‘moderate to high’ tooth wear compared with 60% from OSBG and 35% of individuals from St Mary’s (Table 10). Evidence of ‘pipe facets’ associated with long-term pipe smoking was seen in all seven of the New Zealand adults (sex unknown), two adult females from the OSBG sample, and three adult males from St Mary’s (Figure 2). No information was available for the category of tooth wear from the Cadia or Cross Bones cemeteries.

St Mary’s and Cross Bones had the same percentage of individuals with carious lesions present (53%) (Table 10). Cadia Cemetery had the smallest percentage of individuals with caries present and St John’s had the highest (Table 10). Periodontal disease was identified in more individuals from St John’s, NZ (100% of the sample) and Cross Bones, UK (58%) than in the St Mary’s sample (23%) (Table 8). Data for this category were not available from Cadia Cemetery or the OSBG in NSW.

Evidence of one periapical lesion or more was seen in 18% of individuals from the Cross Bones Burial Ground compared with 5% and 3% of people from the Cadia and St Mary’s samples, respectively (Table 10). St John’s and the OSBG did not have data for this category. Enamel hypoplastic (EH) defects were identified in individuals from four of the five cemeteries (Table 10). Cadia Cemetery did not have specific data for this category. Statistical analyses carried out for the comparison of St Mary’s findings with other samples used the standard error and confidence interval for each percentage.

## 4. Discussion

### 4.1. Multiple Methods for Additional Data

The validity and reproducibility of the methods used here for the investigation of bone and dental samples using LV and SV Micro-CT systems has been established [21]. The methods used in this study are non-invasive, as required for rare archaeological specimens such as the St Mary’s sample. The 2D and 3D high-resolution images were produced from the reconstructed LV and SV Micro-CT scan data sets. These provided information that could not be obtained via the standard macroscopic and radiological investigations (Figure 5) [21]. For large specimens such as the dentoalveolar complex in situ within the human skull where minimal handling is required, the LV Micro-CT scanner is ideal. The SV Micro-CT is suitable for small specimens such as isolated individual teeth. The resolution of the Micro-CT scanning systems depends on the size of the specimen, i.e., the smaller the specimen, the higher the available resolution, and vice versa.

### 4.2. Aim 1—The Oral Health Status of St Mary’s Cemetery Sample

There is a greater precision available with infants and subadults than with adults in assessing dental age based on the eruption and development of teeth [5,6]. Dental age and skeletal age are two different parameters that often do not coincide in an individual. For two subadults (SMB 28, SMB 70) the dental age range estimated by the London Atlas [5,6] using dental radiographs showed some variation from the assigned skeletal age range (Appendix A). These subadults all had comorbidities (Table 6), which could have affected the timing of their dental development and eruption [61,62,63].

Verma et al. (2022) [28,64] reviewed multiple methods that can be used for the determination of the age of adults using dentition. Many of these methods, however, are destructive. The non-invasive methods set out by Miles (1962) and Molnar (1971) [29] and others like them, such as Smith (1984) [57], Littleton and Frochlich (1993) [58], and Scott (1979) [59], assess and score tooth wear (attrition) on the incisal/occlusal surface. These studies can give an indication of both the functional age of the teeth and a predicted age for the individual [28] as well as the severity of tooth wear [29]. Since tooth wear increases with age, older individuals in general will have had more wear. However, differences that were seen in the dental ages of the investigated adults from St Mary’s using Miles’s (1962) [28] system compared with their skeletal age range (Appendix A) could be due to loss of molars in life. The scoring of the functional age of a tooth from tooth wear will be affected if it does not have an opposing molar and will thus affect the predicted age given for an individual.

The tooth wear scores for all tooth types [29] for the St Mary’s sample showed that various teeth (i.e., central and/or lateral incisors and/or canines) had different degrees of wear (Table 2). This may have resulted from differences in diet and/or specific habits and/or the presence or absence of opposing teeth due to antemortem tooth loss. A specific example of the sort of diet that the St Mary’s individuals may have had comes from a 19^th^-century coroner’s report of a man who fell under the wheels of a bullock cart. The “deceased was sitting on the load, which consisted of tea/sugar and other stores” [65] after an annual trip to the City of Adelaide. A witness to the accident observed that “there was about a ton and a half in the dray”, enough supplies for the extended family of the man in question [65].

#### 4.2.1. Dental Pathologies

Loss of teeth in previous generations may have occurred due to poor oral health, such as extensive caries and/or periodontal disease [42,66]. In addition, due to the high cost of dental treatment, some people may have opted for the extraction of teeth in place of preventative and restorative dental treatments [67]. In some cultures and social classes, the tradition of removing a future bride’s teeth and replacing them with dentures as a prenuptial gift was practiced [68,69]. This voluntary edentulism was to prevent the burden of costly dental treatments for the new husband’s family. One adult female (SMB 84) was buried at St Mary’s ‘free ground’ area with a full set of vulcanite and porcelain dentures (Figure 1). The background of this individual was unknown, but they appear to have had a downturn in their economic status before death, as the burial was at the expense of the South Australian Government.

Carious lesions were identified in more than half of the individuals from the St Mary’s sample (*n* = 21/40—Table 3), the majority of whom were adults (*n* = 17/21—Table 3). Taking into account antemortem tooth loss that was mainly due to caries, notwithstanding non-clinical reasons, more teeth/individuals in this sample would have been affected. Most of the carious lesions were located *above or on* the cementoenamel junction (CEJ), rather than below (Appendix A). A ‘build-up’ of dental plaque [3,38] within the interproximal area of the teeth may explain why a greater number of carious lesions were recorded on the mesial and distal surfaces of the teeth remaining in the jaw for the St Mary’s individuals (Appendix A). Extensive carious lesions on other surfaces of the tooth, such as the occlusal surface, may have resulted in its extraction. Calculus deposits seen on the tooth enamel at these locations for many adults from St Mary’s (Table 4 and Appendix A) indicate the accumulation of plaque that had contained caries-producing bacteria, which was then calcified [66,70]. This is also indicative of a cariogenic diet high in carbohydrates and sugars, such as the consumption of bread made of refined flour and tea with sugar [71,72,73,74,75]. Inclusion of such dietary factors as described in the afore mentioned 1859 coroner’s report [65] could have resulted in the high incidence of caries in the St Mary’s sample. The ‘Adelaide Commercial Report’ for July 1853, published in a local newspaper [76], states that “business was exceedingly flat…for all description of goods, both colonial and imported, but tea, sugar and brandy were the exceptions, all of which are in demand at full rates”.

A lack of knowledge and understanding of the effects of diet on oral hygiene [77,78] and the poor availability and/or affordability of dental services [8] could have also contributed to the poor oral health of several individuals from St Mary’s. The periapical cavity (Figure 3) [79] surrounding the roots of the lower-right first permanent molar of subadult SMB 79, with an opening in the buccal surface of the alveolar bone, would have arisen from a large carious lesion that extended into the pulp of the tooth.

The 11 adults (*n*-11/18 adults in the sample) with horizontal bone loss (4 mm or more) from the CEJ to the crest of the alveolar bone and the 12 adults with morphological changes in the alveolar bone (grade 3 or above) [42] suffered from periodontal disease. Six individuals (SMB 14, SMB 53C, SMB 61, SMB 63, SMB 78, and SMB 85) with higher scores for these categories also had extreme antemortem tooth loss (Table 4), which is often seen with advanced periodontal disease.

Lifestyle habits such as smoking (Figure 2) have been linked to an increased risk of periodontal disease [80,81,82,83,84]. New migrants to the colony were encouraged to take up tobacco smoking, by advising them that “if they were in the bush, hungry, thirsty or tired” smoking would remedy the situation [85]. Tooth wear patterns indicating long-term pipe smoking were seen in three adult males (SMB 59, SMB 72, and SMB 78) from the St Mary’s sample. Two of these individuals also had evidence of periodontal disease (Table 6).

#### 4.2.2. Developmental Dental Defects

Extensive evidence of enamel hypoplastic (EH) defects was seen on the dentitions of 14 adults and 10 subadults (*n* = 24/40) (60%) (Table 5 and Appendix A). Eight individuals (four adults, four subadults) from this group had EH defects on more than 60% of their teeth, with some teeth having multiple defects (Table 5 and Appendix A). The presence of multiple EH defects on the teeth indicates that the individuals suffered repeated chronic health insults, which affected the developing teeth [5,6]. Recovery from these episodes of ill health in the colony, during voyaging, or before migration did occur as many of the individuals with EH defects in the St Mary’s sample were adults (Table 5 and Appendix A). The canine teeth of these individuals were more affected by EH defects compared with the incisors, premolars, or molars (Table 6 and Appendix A), suggesting that many health insults occurred in childhood during the development of the primary and/or permanent canines, rather than in young adulthood [5,6].

Limited access to or the affordability of health services could have extended an illness. Infant SMB 58 and subadult SMB 70 both showed extensive enamel hypoplastic defects on both the primary and permanent dentitions [19,56,86,87] (Table 5 and Appendix A). The location of the EH defects on the erupted primary and developing permanent teeth of infant SMB 58 (Figure 4) (Table 5 and Appendix A) suggests that this infant suffered health insults around the time of birth as well as postnatally [4]. The pattern of enamel defects seen on the primary teeth of SMB 58 matches those found by Fearne et al. (1990) [88] in low-birth-weight children.

Dentine, like enamel, could be affected by health insults during dental development. Areas of defective mineralisation in the dentine are referred to as interglobular dentine (IGD) [4]. Such defects can occur in the same tooth in addition to EH defects [1,89,90]. Furthermore, these internal dentine defects can be seen at a similar level to the external EH defect (Figure 5) [1,91], suggesting that the same systemic health insult caused both defects. Areas of IGD and EH defects at a similar level in the tooth were seen in four individuals (one infant, SMB 58; one subadult, SMB 70; and two adults, SMB 63 and SMB 73 (Table 7 and Figure 4 and Figure 5). No one in the St Mary’s sample was observed with IGD alone. The clinical manifestations of dental defects are related to the severity and duration of the health insult and the degree of the host’s response [92].

### 4.3. Aim 2: Oral Health Conditions and General Health Status

Evaluation of the oral health findings together with information regarding general health from the skeletal evidence for St Mary’s sample [1,2] provided additional insight into the overall health and lives of these early colonists. Seven adults from this sample had less than ten teeth present, while four individuals in this group (SMB 14, SMB 63, SMB 78, and SMB 85) had just two to four teeth each (Table 3, Table 7 and Appendix A). This antemortem tooth loss indicates that their masticatory function was substantially reduced. Dental clinicians have indicated that “20 teeth, with nine to ten pairs of contacting units are necessary to maintain adequate masticatory efficiency” [93]. Therefore, many of the adults in the St Mary’s sample would not have had a sufficient number of teeth to maintain adequate masticatory efficiency required for good general health and wellbeing [94].

Seven of the St Mary’s adults had evidence of vertebral osteophytes, eburnation of the vertebral articular facets and other joints of the body (Table 7) [2]. These changes to the bones due to joint disease could have limited their dexterity and the ability to maintain good oral hygiene [10]. It has been established that there is a relationship between poor oral health and systemic disease [9,11,12,13,14,16,17,95,96]. Individuals from St Mary’s showed extensive evidence of poor oral health such as carious lesions, periodontal disease, and antemortem tooth loss (Table 3, Table 7 and Appendix A), which would have affected their general health.

The St Mary’s adults with extensive periodontal disease did not show evidence of pathological manifestations associated with systemic disease on their skeletons (Table 7). A lack of skeletal signs of chronic ill health near the time of death does not necessarily indicate an absence of comorbidities, or indeed a sign of previous good general health. The extensive dental disease seen in many of the St Mary’s individuals could have increased the severity of some systemic health conditions such as atherosclerotic cardiovascular disease [12,13,14,15]. This cardiovascular disease only affects the soft tissues of the body and does not leave pathological manifestations on the skeleton [12,13,14,15], while others, for e.g., diabetes mellitus, can involve the skeleton [97,98,99]. However, the signs in the skeletal material may not be pathognomic, allowing for the identification of the specific aetiology.

Some chronic health insults suffered during development, for example nutritional deficiencies and/or infectious diseases such as treponemal disease (syphilis—congenital or acquired) and/or tuberculosis could produce dental defects (i.e., EH and IGD) as well as skeletal manifestations [1,100,101,102,103]. Twelve individuals from St Mary’s had both developmental dental defects and skeletal signs of a comorbidity present (Table 7). The presence of specific EH defects with subadult SMB 70, (dental age between 11.5–12.5 years ±1 year) as well as IGD, together with evidence of the involvement of the skeleton, including an “osteoblastic lesion on the cranial vault”, pathological changes to the clavicle, ribs, and vertebrae (Table 7) [56], all indicated that this subadult had suffered from congenital syphilis, tuberculosis, and mercurial toxicity [19,56,87]. This is an example of the multiple interactions between factors in oral health and in general health which are components in a multilevel complex interactive network that operated throughout the lives of these individuals.

### 4.4. Aim 3: St Mary’s Oral Health Findings Compared with Australian, New Zealand, and British Historic Samples

Individuals from the 19th century in Cadia Cemetery, NSW, Australia [52] and St John’s Burial Ground, Milton, Otago, New Zealand [54], represented contemporary communities (Table 8 and Table 9) that could have had similar lifestyles and occupations (agricultural and industrial—mining, etc.) to the individuals buried at the St Mary’s Cemetery ‘free ground’ (Table 8 and Table 9). Individuals interred at the Old Sydney Burial Ground (OSBG), NSW [53], were not contemporaries of the St Mary’s sample and could have come from different backgrounds, i.e., they could have been convicts or serving as sailors in comparison to the free settlers in South Australia. They could have also lived in different environmental conditions. However, as previously stated, there are only a few colonial Australian skeletal samples, and it was thought a worthwhile comparison. Cross Bones Burial Ground, London, UK, [55], was used for the poorest people in the 19th century Southwark community [104]. Individuals interred here did not have the money to pay for a burial or a headstone memorial, similar to those buried in the unmarked ‘free ground’ section at St Mary’s Church Cemetery.

The outcome of the comparison of data between these historic cemeteries for five oral health categories found similarities and differences. Information on tooth wear was limited to the OSBG and St John’s samples. Compared with these samples, there seems to be only a small number of individuals from St Mary’s with ‘moderate to high’ wear on the occlusal surfaces of the dentition (Table 10). This may be due to a difference in diet, i.e., the individuals from NSW and NZ probably had a more abrasive diet. However, the numbers in the OSBG and St John’s samples are small and therefore when statistics were applied, there was considerable overlap of confidence levels and no statistical significance.

Similarities were seen between St Mary’s and Cross Bones Burial Ground (UK) for the categories of carious lesions and enamel hypoplastic (EH) defects (Table 10). In relation to developmental enamel defects, both the St John’s and OSBG cemeteries had more individuals with EH present than the St Mary’s or Cross Bones cemeteries (Table 10). Brook and Smith (1998) [105] investigated developmental defects in a 20th-century sample of East London school children. They found that 14.6% of the sample had enamel hypoplasia [105], which is markedly lower than any of the historic cemeteries compared in this study.

The same percentage of individuals with carious lesions (53%) for St Mary’s and Cross Bones (Table 10) could suggest that the background of the early settlers to South Australia was similar to those buried at Cross Bones and that they continued with their oral hygiene and dietary habits.

Periodontal disease had a higher frequency in the British and New Zealand samples than St Mary’s (Table 10). The individuals from St John’s were all adults compared with St Mary’s, which had a higher number of subadults (Table 8). Cross Bones had a similar percentage of subadults to the St Mary’s sample, suggesting that many of the St Mary’s individuals maintained a better standard of oral hygiene than the individuals from the British sample.

In summary, considering the limitations of the sample sizes and variations in the presentation and availability of data for analysis, the oral health of many individuals in the comparison samples was poor (Table 10), which would have affected their general health. The seven individuals from St John’s Cemetery sample, NZ, appear to have had the worst oral health compared with data from the other historic samples, but the small sample size from this cemetery could have influenced these findings (Table 10). The oral health of individuals from St Mary’s was poor but was better than the other cemetery samples for many of the oral health categories scored in this study (Table 10).

### 4.5. Limitations of This Study

The use of multiple non-invasive methods allowed detailed data to be collected. While the LV Micro-CT technique is ideal for high-resolution analysis of dentition in situ within archaeological human skull samples [21], the cost of this micro-CT system limited the number of individuals from the St Mary’s sample that were scanned. The number of isolated individual tooth samples available for SV Micro-CT scanning from the St Mary’s sample was also a limitation. As this equipment only scans small objects, teeth have to be separated from the jaw, and so destructive analysis was prohibited. The SV Micro-CT scanning method was included in this study even though the information obtained from an individual tooth is restricted. These data cannot infer the overall oral or general health of a person, but they can provide information on external and internal developmental defects.

Miles (1962) [28] acknowledges that the method of assessing the functional age of a tooth by tooth wear patterns, and thereby predicting the age of the subject, has limitations. The individuals studied [28] were derived from a different population (Anglo Saxon). These people could have had a different diet and a different caries rate to the 19th-century group, as they would have had less processed sugar. A lower caries rate meant less antemortem tooth loss, and so they could have retained their permanent teeth longer when compared with the adults from the St Mary’s sample. A limitation that could have affected the results of the presence, location, and severity of calculus deposits [31,32,33] on the dentitions of the St Mary’s adults was due to the partial removal of calculus from some of these skulls for previous studies [106].

## 5. Conclusions

The overall oral health of the settlers buried at the St Mary’s Cemetery ‘free ground’ area was poor. Their inability to consume adequate amounts of nutritious foods due to extensive antemortem tooth loss would have exacerbated comorbidities and impacted their general health. The oral health of St Mary’s settlers in relation to categories of tooth wear, periodontal disease, and periapical abscess was better than individuals from the comparison Australian, New Zealand, and British cemeteries. The St Mary’s sample had similar findings for caries and enamel hypoplasia to the Cross Bones Burial Ground in London, suggesting that little improvement had occurred since arriving in the new colony. A high percentage of individuals from four of the five historic cemeteries (St Mary’s; Old Sydney Burial Ground, NSW; St John’s, NZ; and Cross Bones Burial Ground, UK) had enamel hypoplastic defects. This indicates that these individuals had suffered systemic health insults during dental development, which could have been commonplace during the 19th century.

## Figures and Tables

**Figure 1 dentistry-11-00099-f001:**
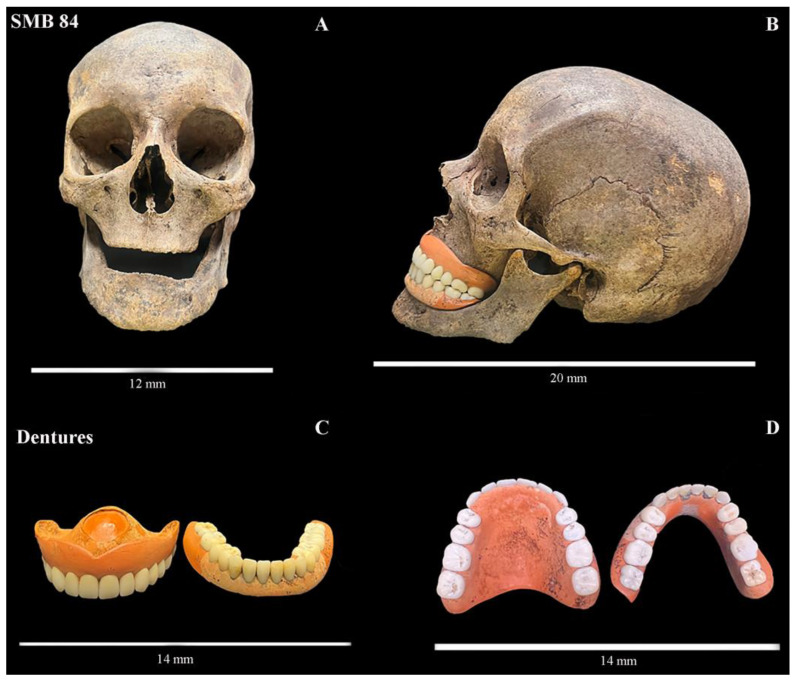
SMB 84—adult female (age range 50–59 years.). Vulcanite base dentures with porcelain teeth. Images show dentures in situ and removed from the maxilla and mandible. (**A**). Anterior view of the edentulous maxilla and mandible. (**B**). Left lateral view of the ‘well-made’ dentures with a class one occlusion in the jaw. (**C**,**D**). The vulcanite dentures with porcelain teeth—the maxillary denture shows a shield shape suction area on the superior surface of the plate. © Angela Gurr.

**Figure 2 dentistry-11-00099-f002:**
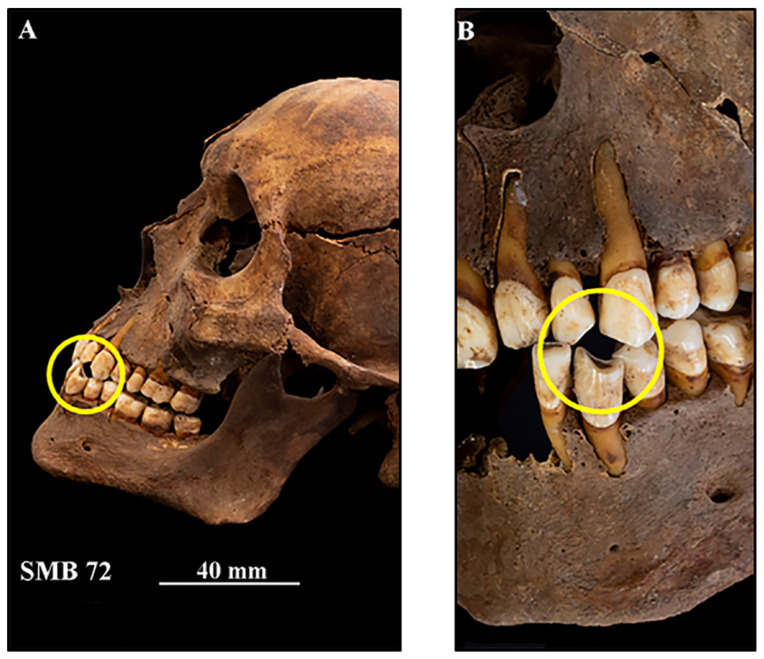
SMB 72. Adult male (age range 40–49 years). (**A**): Left lateral view of the dentition in situ in the skull. Yellow circle highlights a pattern of tooth wear often referred to as a ‘pipe notch’. The teeth have become worn from the movement of and holding a smoking pipe between the teeth over an extended period of time. (**B**): A close-up view of the left maxillary and mandibular permanent lateral incisors and canines opposing each other. The yellow circle shows a clear semi-circular tooth wear pattern. © Angela Gurr.

**Figure 3 dentistry-11-00099-f003:**
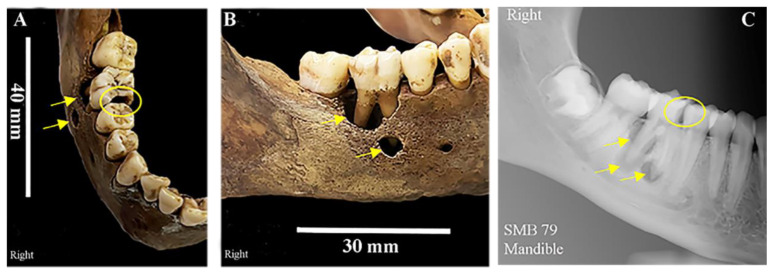
SMB 79, subadult (dental age range 15.5–16.5 years (±1 year)—skeletal assessment age range 16–18 yrs.). (**A**,**B**): Yellow circle shows caries in the mesial occlusal surface of the lower right permanent first molar (M1—FDI number 46). Yellow arrows indicate the periapical cavity and opening in the buccal surface of the alveolar bone. (**C**): Periapical dental radiograph of the mandible—yellow arrows show the extent of the cavity surrounding the roots of the M1 molar, and the ‘opening’ in the alveolar bone adjacent to the apex of the mesial root. The yellow circle highlights caries that extended to involve the pulp. © Angela Gurr.

**Figure 4 dentistry-11-00099-f004:**
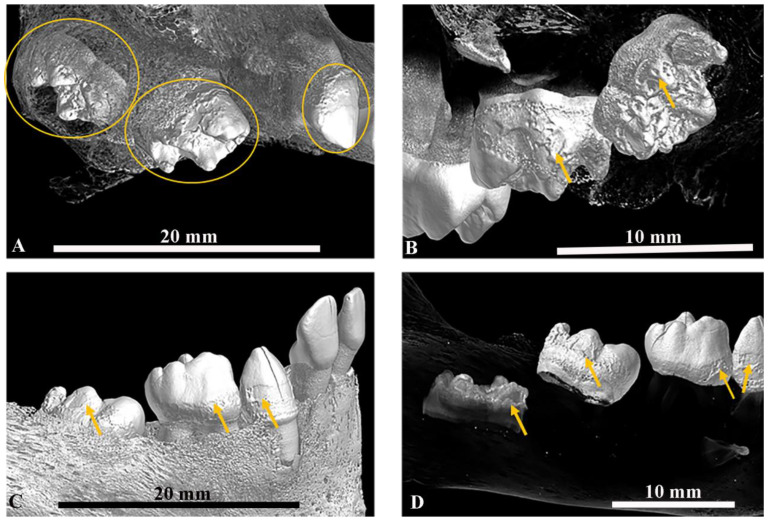
SMB 58—Infant (dental age range between 1–1.5 ± 3 months, skeletal assessment age range between 0–2. years of age). (Cranium—Large Volume Micro-CT scanned at 35 µm/pixel and mandible at 21 µm/pixel, Appendix A). (**A**,**B**,**D**): Created using Avizo 9 software [30]; the bone density threshold level was manipulated/reduced to reveal the denser structures of the developing teeth within the alveolar bone. (**A**): Buccal view of the semi-erupted, upper-right primary canine and primary second molar; the first primary molar was lost post-mortem. Yellow circles show teeth affected by enamel hypoplastic defects, including the developing crown of the permanent upper right first molar within the alveolar bone. (**B**): Lingual/palatal view of the upper-left primary first and second molars and the crown of the permanent first molar within the alveoli. Yellow arrows highlight enamel hypoplastic defects. (**C**). Lateral view of the mandibular primary dentition. Yellow arrows indicate some enamel hypoplastic defects on the primary canine, first molar, and erupting second molar. (**D**): As with image B, the bone density threshold level was changed to show the developing teeth within the alveoli—yellow arrows show the enamel hypoplastic defects of the erupted primary teeth, semi-erupted primary second molar, and the cusps of the developing crown of the permanent first molar. © Angela Gurr.

**Figure 5 dentistry-11-00099-f005:**
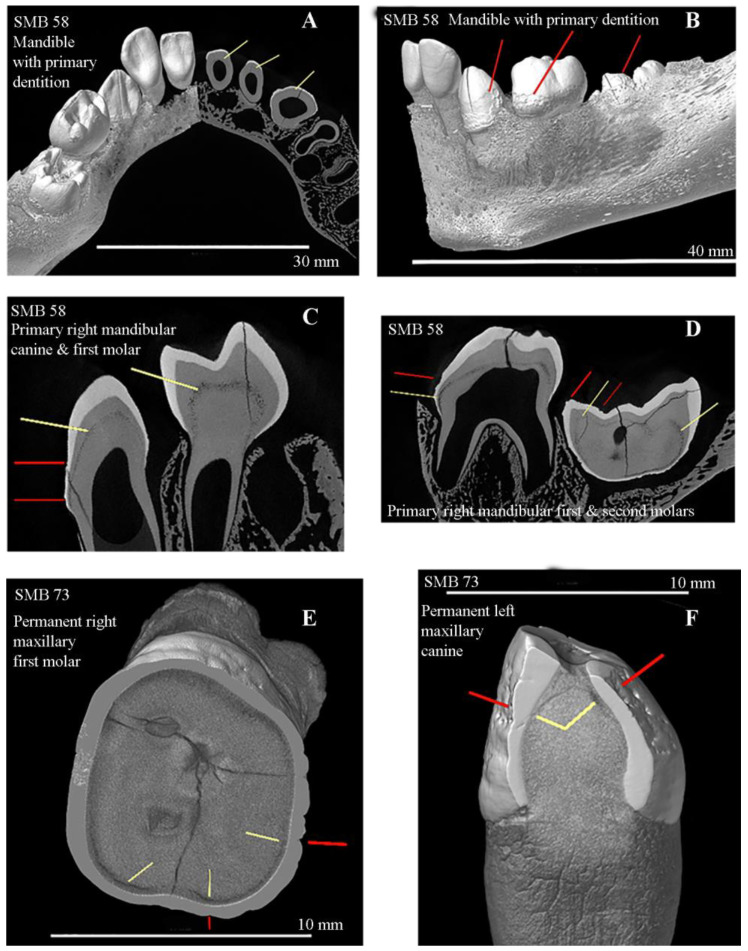
(**A**–**D**): SMB 58—infant (dental age range between 1–1.5 ± 3 months, skeletal age range between 0–2 years of age). Large Volume (LV) Micro-CT scanned at 21 µm/pixel (Appendix A). (**A**): Superior/posterior view of the mandible and primary teeth. The left side of the jaw and teeth is shown in 3D and the right side is a 2D LV micro-CT slice of the teeth—yellow lines show interglobular dentine (IGD). (**B**): Lateral view of the mandibular primary dentition—red lines show enamel hypoplastic (EH) defects. (**C**,**D**): Sagittal slices from LV micro-CT scans of the mandibular dentition (Image C—FDI tooth numbers 83 and 84, Image D—FDI teeth 84 and 85). Yellow lines show areas of IGD in the internal structure of the teeth and the red lines show EH defects on the external structures of the teeth. (**E**,**F**): SMB 73—adult male, (skeletal age range: between 30–39 years). Individual tooth samples scanned with the Small Volume (SV) Micro-CT system at 9 µm/pixel (Appendix A). (**E**): FDI tooth number 16; (**F**): FDI tooth number 23. Both images—yellow lines show areas of IGD and red lines show EH defects in the teeth and at a similar level. All images were created using Avizo software [30] Volren or Orthoslice functions from reconstructed LV and SV Micro-CT scan data. © Angela Gurr.

**Table 2 dentistry-11-00099-t002:** The number of each tooth type for each category of tooth wear scored using Molnar’s system [29] for the permanent dentition of the adults from the St Mary’s sample, with the number of each tooth type for each category of tooth wear scored.

Type of Permanent Tooth	Number of Teeth with Cat. 4	Number of Teeth with Cat. 5	Number of Teeth with Cat. 6	Number of Teeth With Cat. 7	Number of Teeth With Cat. 8	Total Number of Each Tooth Type
**Cent. Incisor**	13	13	0	0	0	26
**Lat. Incisor**	22	18	0	0	0	40
**Canine**	16	14	3	0	0	33
**P1**	8	6	0	1	0	15
**P2**	9	1	2	1	0	13
**M1**	7	1	4	0	0	12
**M2**	7	3	3	1	0	14
**M3**	1	1	3	0	1	6
**Total**	**83**	**57**	**15**	**3**	**0**	

Notes: Cent. = central, Lat. = lateral, P1 = first premolar, P2 = second premolar, M1 = first permanent molar, M2 = second permanent molar, M3 = third permanent molar. Cat. = category of tooth wear.

**Table 3 dentistry-11-00099-t003:** Number and percentage of carious lesions identified for individuals from the St Mary’s sample, with identification of teeth lost in life (antemortem).

St Mary’s ID	Age Range	Sex	Total Number of Teeth Present	Permanent or Primary Dentition	* Total Number of Teeth Affected by Carious Lesions	Percentage of Teeth Affected by Carious Lesions	Total Number of Carious Lesions Present	** Antemortem Tooth Loss FDI Notation Number/s of Teeth Lost in Life
**SMB 19**	6–9	U	22	7 primary 15 Permanent	2	9%	3	None
**SMB 70**	6–9	U	8	2 primary 6 Permanent	2	25%	3	None
**SMB 28**	10–14	U	26	Permanent	1	4%	2	None
**SMB 79**	15–18	U	25	Permanent	8	32%	13	36
**SMB 05**	20–30	F	6	Permanent	4	67%	8	11, 12, 13, 14, 15, 16, 18, 21, 22, 23, 24, 25, 26, 27,28, 36, 37, 38, 45, 46, 47, 48
**SMB 53C**	30–39	F	9	Permanent	8	89%	12	15, 16, 17, 18, 24, 25, 27, 28, 31, 35, 36, 37, 38, 41, 44, 47, 48
**SMB 66B**	30–39	F	17	Permanent	11	65%	16	11 (root only), 13, 16, 18, 26, 28, 36, 37, 38, 46, 47, 48
**SMB 73**	30–39	M	19	Permanent	17	89%	39	15, 17, 18, 24, 26, 27, 36, 37, 38, 46, 47, 48
**SMB 06**	40–49	M	24	Permanent	12	50%	19	28, 34, 35, 36, 38, 48
**SMB 09**	40–49	M	22	Permanent	1	5%	11	11, 27, 38
**SMB 57**	40–49	M	25	Permanent	8	32%	12	25, 28
**SMB 61**	40–49	F	7	Permanent	4	57%	5	11, 12, 15, 16, 17, 18, 21, 22, 24, 26, 27, 28, 334,35, 36, 37,38, 43, 45, 46, 47, 48
**SMB 72**	40–49	M	29	Permanent	7	24%	15	None
**SMB 78**	40–49	M	4	Permanent	2	50%	3	11, 12, 15, 16, 17, 18, 21, 22, 23, 24, 25, 26, 27, 28, 31, 32, 36, 37, 38, 41, 42, 43, 44, 45, 46, 47, 48
**SMB 83**	40–49	M	16	Permanent	6	38%	7	16, 18, 26, 36, 45, 46
**SMB 85**	40–49	M	2	Permanent	1	50%	1	11, 12, 13, 14, 15, 16, 17, 18, 23, 24, 25, 26, 27, 28, 31, 35, 36, 37, 38, 41, 43, 44, 45, 46, 47, 48
**SMB 14**	50–59	M	2	Permanent	1	50%	1	12, 14, 15, 16, 17, 18, 21, 22, 24, 25, 26, 27, 28, 34, 36, 37, 38, 46, 48
**SMB 23**	50–59	M	21	Permanent	20	95%	54	48
**SMB 59**	50–59	M	15	Permanent	10	67%	10	14, 15, 24, 25, 47
**SMB 63**	50–59	M	3	Permanent	1	33%	3	11, 12, 14, 15, 16, 17, 21, 22, 24, 25, 26, 27, 28, 35, 36, 37, 38, 41, 42, 44, 46, 47, 48
**SMB 68**	50–59	M	19	Permanent	11	58%	14	15, 16, 36, 41, 46
**TOTAL**			**321**		**137**		**251**	

Notes: F = female, M = male, U = undetermined. * A tooth could have more than one carious lesion present. ** Antemortem tooth loss = evidence of healing or healed alveolar bone in the location of the absent tooth. FDI = Fédération Dentaire Internationale tooth identification notation system.

**Table 4 dentistry-11-00099-t004:** Calculus—the percentage of individuals and the number of teeth with calculus deposits.

**Dental Pathology:** **Calculus**	**Sample Size** **(Adults Only)** ***n*** **=**	**Percentage of Individuals Affected**	**Number of Teeth Present**	**Number of Teeth with Calculus Deposits**	**Percentage** **of Teeth with Calculus**	**Mean Number** **of Teeth Affected**
	**17**	**65%**	**240**	**79**	**33%**	**7.2**

**Table 5 dentistry-11-00099-t005:** Number and percentage of enamel hypoplastic (EH) defects identified in the individuals from the St Mary’s sample with antemortem tooth loss.

St Mary’s ID	Age range (Skeletal and Eruption Findings) (Years)	Sex	Total Number of Teeth Present	Permanent and/or Primary Dentition	Tooth Type/s Affected by EH Defects FDI Notation	Total Number of Teeth with EH Defects	Percentage of Teeth Affected by EH Defects	Type of EH Defect/s Present
**SMB 58**	0–2	U	11	All primary teeth	51, 52, 54, 61, 62, 72, 74, 84	8	72%	Linear and pits
**SMB 11**	0–2	U	19	primary	53, 63, 73, 83	4	21%	Pits
**SMB 04A**	3–5	U	19	primary	71, 72, 81, 82	4	21%	Pits
**SMB 35**	6–9	U	11	primary	63	1	9%	Pits
**SMB 19**	6–9	U	22	Mixed 7 primary 15 Permanent	12, 13, 14, 15, 21, 22, 42, 83	8	36%	Linear and pit
**SMB 51**	10–15	U	16	1 primary 15 Permanent	12, 13, 14, 16, 22, 23 24, 25, 26, 34, 36	11	69%	Linear and pit
**SMB 52B**	10–15	U	17	2 primary 15 permanent	11, 12, 13, 16, 21, 23, 26, 31,32, 33, 36, 14, 42, 43	14	82%	Linear and pit
**SMB 70**	10–15	U	8	2 primary 6 Permanent	53, 21, 63, 26	4	50%	Linear and pits
**SMB 28**	10–15	U	26	All Permanent	11, 12, 15, 16, 22, 23, 25, 26, 27, 31, 32, 33, 34, 35, 41, 42, 43, 44, 45, 47	20	77%	Linear and pit
**SMB 79**	16–19	U	25	Permanent	13, 23, 27, 33,43	5	20%	Linear and pit
**SMB 05**	20–29	F	6	Permanent	41, 42	2	33%	Linear
**SMB 53C**	30–39	F	9	Permanent	11, 13, 14, 21, 23	5	56%	Linear and pit
**SMB 66B**	30–39	F	17	permanent	14, 17, 21, 23, 27, 34, 44, 45	8	47%	Pits
**SMB 73**	30–39	M	19	Permanent	11, 12, 13, 16, 21, 23, 25, 31, 32, 33, 34, 41, 42, 43, 44	15	79%	Linear and pits
**SMB 06**	40–49	M	24	Permanent	12, 31, 32, 41, 42	5	21%	Linear
**SMB 09**	40–49	M	22	Permanent	12, 13, 21, 22, 23, 32, 33, 43	8	36%	Linear and pit
**SMB 57**	40–49	M	25	Permanent	11, 12, 13, 14, 27, 32, 33, 35, 42, 43	10	40%	Linear and pit
**SMB 72**	40–49	M	29	Permanent	12, 13, 23, 24, 38, 43, 48	7	24%	Pits
**SMB 83**	40–49	M	16	Permanent	13, 21, 23, 27, 28, 33, 34, 42, 43	9	56%	Linear and pit
**SMB 85**	40–49	M	2	Permanent	33	1	50%	Linear and pit 2
**SMB 23**	50–59	M	21	Permanent	17, 18, 21, 31, 32, 33, 41, 42, 43	9	43%	Linear and pit
**SMB 59**	50–59	M	15	Permanent	11, 12, 13, 21, 22, 23, 35, 38, 41, 42, 44, 46	12	80%	Linear and pit
**SMB 63**	50–59	M	3	Permanent	32, 43	2	67%	Pits
**SMB 68**	50–59	M	19	Permanent	14, 23, 35, 43, 44, 45	6	42%	Pits

Notes: M = male, F = female, U = undetermined sex. EH = enamel hypoplastic defects. A tooth may have more than one EH defect. FDI = Fédération Dentaire Internationale tooth identification notation system.

**Table 6 dentistry-11-00099-t006:** The different tooth types affected by enamel hypoplastic defects in the St Mary’s Cemetery sample.

Tooth Type	Number of Primary Teeth with EH Defects	Number of Permanent Teeth with EH Defects	Total Number of Each Tooth Type
**Cent. Incisor**	4	32	**36**
**Lat. Incisor**	5	29	**34**
**Canine**	8	44	**52**
**P1**	n/a	18	**18**
**P2**	n/a	12	**12**
**M1**	3	10	**13**
**M2**	0	8	**8**
**M3**	n/a	5	**5**
**Total**	20	158	

Notes: EH = Enamel hypoplastic defects. Cent. = central, Lat. = lateral, P1 = first premolar, P2 = second premolar, M1 = first molar, M2 = second molar, M3 = third permanent molar.

**Table 7 dentistry-11-00099-t007:** Key individuals from the St Mary’s Cemetery sample with a summary of their oral health findings. Many individuals had one or more skeletal signs of a comorbidity and/or signs of skeletal and dental changes.

St Mary’s Burial I/D	Sex	Dental Age D = Skeletal Age S = (Years)	Inventory Number of Teeth Present de = Deciduous P = Permanent	Caries Number of Teeth Affected	Periodontal Disease		EH Number of Teeth Affected † Max:	IGD Number of Teeth Affected	Comorbidities Signs of Skeletal and Dental Changes
					Alveolar Bone Status Grade 1–4 Min–Max	Alveolar Bone Loss Number of Teeth Affected			
**SMB 58**	**U**	**D= 1–1.5 ** (**±3 months**) **S= 0–2**	**10 de**	**0**	**1**	**0**	9/10	10/10	(i) Abnormal porosity of the cortical bones of the maxilla and mandible [1].
**SMB 04A**	U	D= 3.5–4.5 (±6 months) S = 2–4	19 de	0	1	0	4/19	0	(i) Cribra orbitalia Types 3–4 [1].
**SMB 19**	U	D = 7.5–8.5 (±1 year) S = 5–9	7 de 12 P	2/19	1–2	0	8/19	0	(i) Cribra orbitalia Types 3–4 [1].
**SMB 70**	U	D = 11.5–12.5 (±1 year) S = 8–9	2 de 6 P	6/8	1–2	0	4/8	2/2	(i) Congenital Syphilis, (ii) TB, (iii) mercury toxicity, (iv) abnormal porosity in the cortical bones of the greater wing of sphenoid, maxilla, scapulae, pelvic bones, bilaterally [1,19,56].
**SMB 52 B**	U	D = 10.5–11.5 (±1 year) S = 8–12	2 de 14 P	0	1	1/16	14/15	0	None seen.
**SMB 51**	U	D = 10–11. (±1 year) S = 8–12	14 de	0	1–3	0	10/14	0	None seen.
**SMB 28**	F	D = 15.5–16.5 (±1 year) S = 10–14	26 P	1/26	1–4	0	20/26	0	(i) Cribra orbitalia Type 4, (ii) Possible nutritional deficiency due to abnormal porosity of the cortical bones of the greater wing of the sphenoid and alveolar tissue of the maxilla—bilaterally [1].
**SMB 79**	U	D = 15–16 (±1 year) S = 16–18	25 P	13/25	1–2	0	5/25	0	(i) Spina bifida occulta, (ii) evidence of a dental abscess (Figure 3).
**SMB 05**	F	D = over 23.5 S = 20–29	6 P	4/6	1–2	0	2/5	0	None seen.
**SMB 53 C**	F	D = over 23.5 S = 30–39	9 P	8/9	2–3	8/9	5/9	0	(i) Pitting of the external surface of the Occipital bone, (ii) several vertebral osteophytes- [2].
**SMB 66 B**	F	D = over 23.5 S = 30–39	17 P	12/17	1–4	12/17	8/17	0	None seen.
**SMB 73**	M	D = over 23.5 S = 30–39	19 P	17/19	1–3	15/19	18/19	7/19	(i) Torticollis, (iii) spina bifida occulta.
**SMB 06**	M	D = over 23.5 S = 40–49	24 P	12/24	1–4	22/24	5/24	0	(i) Posterior external surfaces of parietal bones—uneven thickening, possible mild caries sicca. (ii) Vertebral osteophytes and Schmorl’s nodes [1,2,19].
**SMB 09**	M	D = over 23.5 S = 40–49	21 P	11/21	2–4	6/21	8/21	Not micro-CT scanned	(i) Spina bifida occulta.
**SMB 57**	M	D = over 23.5 S = 40–49	25 P	8/25	2–4	21/25	11/25	Not micro-CT scanned	(i) Vertebral osteophytes [2].
**SMB 61**	F	D = over 23.5 S = 40–49	6 P	4/6	2–3	3/6	0	Not micro-CT scanned	(i) Spina bifida occulta.
**SMB 72**	M	D = over 23.5 S = 40–49	31 P	7/31	2–3	2/31	0	0	(i) Evidence of pipe smoker’s tooth wear pattern *. (Figure 2).
**SMB 78**	M	D = over 23.5 S = 40–49	4 P	2/4	0	4/4	0	Not micro-CT scanned	(ii) Vertebral osteophytes, (ii) bony growth 20 mm × 10 mm left fibular ossified haemorrhage [2].
**SMB 83**	M	D = over 23.5 S = 40–49	16 P	6/16	1–3	0	9/16	0	(i) Spina bifida occulta, (ii) vertebral osteophytes, (iii) bony thickening mid-shaft femur 20 mm × 3 mm, healed trauma antemortem [2].
**SMB 85**	M	D = over 23.5 S = 40–49	2 P	1/2	3–4	2/2	1/2	0	(i) Vertebral osteophytes, (ii) eburnation of multiple vertebral facet joints. [2]
**SMB 14**	M	D = over 23.5 S = 50–59	2 P	1/2	2–3	2/2	0	Not micro-CT scanned	(i) Multiple vertebral osteophytes, (ii) eburnation of multiple vertebral facet joints, the joints of the ulna, trochlear, and olecranon (bilaterally), femoral head, acetabulum, talus (head), and the navicular [2].
**SMB 23**	M	D = over 23.5 S = 50–59	21 P	20/21	3–4	10/21	9/21	0	None seen.
**SMB 59**	M	D = over 23.5 S = 50–59	16 P	10/16	3–4	8/16	12/16	Not micro-CT scanned	(i) Evidence of pipe smoker’s tooth wear pattern. (Figure 2).
**SMB 63**	M	D = over 23.5 S = 50–59	3 P	3/3	4	3/5	2/3	1/1	(i) Spina bifida occulta, (ii) concaved maxillary sinus with signs of new bone growth.
**SMB 68**	M	D = over 23.5 S = 50–59	19 P	11/19	2–4	15/19	8/19	0	(i) Eburnation of femoral head, acetabulum, talus, and calcaneus [2].

Notes: D = Dental age range, estimated using the London Atlas of Human Tooth Development and Eruption [5,6]. S = skeletal age range estimated from the assessment of skeletal changes/maturity [19]; U = undetermined sex, M = male, F = female. Periodontal disease includes (i) Alveolar Bone Status—Ogden’s (2008) [42] screening system assigns a grade in relation to the condition of the buccal contours of the alveolar margins of the posterior teeth. A higher grade implies a more severe grade of periodontal disease; grades are between 0–4 (0 = unable to score, 1 = no disease, and 4 = severe periodontitis). Additionally, (2) horizontal alveolar bone loss measurements—the average distance measurement from the CEJ on a tooth to the crest of the alveolar bone that has not been affected by periodontal disease is ~2 mm. The higher distance measurement over this point suggests that horizontal alveolar bone loss has occurred. Measurements of 4 mm and over are included in Table 4. EH = Enamel hypoplasia; a category is assigned to each tooth for the presence or absence of EH defects. Table 4 shows the maximum number of EH defects identified. IGD = Interglobular Dentine—a mineralisation defect in the dentine of the tooth. A category was assigned to each tooth for the presences of this defect. Pipe notch/facet: a tooth wear pattern caused by a pipe stem held and moved between the maxillary and mandibular teeth—indicator of smoking. † A tooth may have more than one EH defect.

**Table 8 dentistry-11-00099-t008:** Demographic profiles of the subadults from St Mary’s Cemetery, SA; Cadia Cemetery, NSW; Old Sydney Burial Ground, NSW; St Johns Burial Ground, NZ; and Cross Bones Burial Ground, UK.

Cemetery:	Total Dental Sample/Total Sample Size	Total Number of Subadults in Dental Sample	Preterm <37 Weeks	Foetal 40 Weeks Post- Partum	Infant 0–11 Months	Subadult 1–5 Years	Subadult 6–11 Years	Adolescent 12–17 Years	Subadult of Unknown Age
**St Mary’s Cemetery** (SA) 1847–1927	40/70	**22/40** **55%**	0	0	4	10	6	2	0
**Cadia Cemetery** (NSW)1864–1927	109/111	**73/109** **67%**	0	15	25	17	3	4	9
**Old Sydney Burial Ground** (NSW)1792–1820	10/10	**0**	0	0	0	0	0	0	0
**St John’s ** **Burial Ground ** (NZ)1860–1926	7/27	**0**	0	0	0	0	0	0	0
**Cross Bones Burial Ground ** (UK)1800–1853	83/147	**39/79** **47%**	0	0	8	26	2	1	2

**Table 9 dentistry-11-00099-t009:** Demographic profiles of the adults from St Mary’s Cemetery, SA; Cadia Cemetery, NSW; Old Sydney Burial Ground, NSW; St Johns Burial Ground, NZ; and Cross Bones Burial Ground, UK.

Cemetery:	Total Dental Sample/Total Sample Size	Total Number of Adults	Age Group 18–22 Years	Young Adult 23–35 Years	Middle Adult 35–50 Years	Old Adult 50+ Years	Adults of Unknown Age	Adult Male	Adult Female	Adults of Unknown Sex
**St Mary’s Cemetery** (SA)1847–1927	40/70	**18/40** **45%**	1	3	8	6	0	13	5	0
**Cadia Cemetery** (NSW)1864–1927	109/111	**36/109** **33%**	0	7	18	10	1	23	14	0
**Old Sydney Burial Ground** (NSW)1792–1820	10/10	**10/10** **100%**	N/A	N/A	N/A	N/A	10	0	4	6
**St John’s Burial Ground** (NZ)1860–1926	7/27	**7/7** **100%**	0	0	4	4	2	4	3	3
**Cross Bones Burial Ground** (UK)1800–1853	83/147	**44/83** **53%**	3	4	18	14	5	12	27	5

**Table 10 dentistry-11-00099-t010:** Comparison of findings for the oral health categories investigated between the St Mary’s Cemetery sample and other 19th-century Australian, New Zealand, and British cemetery samples. The results presented in Table 8 are the number of individuals affected/number of individuals in the total sample for each cemetery, i.e., St Mary’s *n* =/*n* = followed by the percentage of affected individuals from that cemetery.

Cemetery:	Sample Size (Total Number of Individuals) *n* =	Dental and Alveolar Bone Health Categories Scored				
		Tooth Wear ‡ Moderate to Heavy	Carious Lesions Present	Periodontal Disease Interdental Alveolar Resorption	Periapical Abscess Present	Enamel Hypoplastic Defects Present
**St Mary’s Cemetery** **(SA) 1847–1927**	**40**	14/40 35%	21/40 53%	9/40 23%	1/40 3%	24/40 60%
**Cadia Cemetery ** (NSW) 1864–1927	**109**	Results not available	32/109 29%	Results not available	5/109 5%	Results not available
**Old Sydney Burial Ground ** (NSW) 1792–1820	**10**	6/10 60%	4/10 40%	Results not available	Results not available	7/10 70%
**St John’s Burial Ground ** (NZ) 1860–1926	**7**	7/7 100%	6/7 86%	7/7 100%	5/7 71%	6/7 86%
**Cross Bones Burial Ground ** (UK) 1800–1853	**83**	Results not available	44/83 53%	42/83 51%	15/83 18%	48/83 58%

Notes: ooth Wear ‡ Moderate to Heavy—This term was used as each skeletal sample used a different system to score tooth wear, i.e., for the St Mary’s sample, tooth wear was scored using Molnar’s (1971) [29] system; moderate-to-heavy = Category 5 and above; Cadia Cemetery was scored using Smith’s (1984) [57] system; moderate-to-heavy wear = Category 5 and above; Old Sydney Burial Ground was scored using Littleton and Frohlich (1993) [58] and Scott’s (1979) [59] systems; moderate-to-heavy wear = Category 7 or above; Cross Bones Burial Ground was scored for tooth wear using Buikstra and Ubelaker (1994) [60]; moderate-to-heavy wear = Category 5 and above. SA = South Australia, NSW = New South Wales, NZ = New Zealand, UK = United Kingdom.

## Data Availability

The data presented in this study are openly available in Preprints.org https://doi.org/10.20944/preprints202302.0167.v1.

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
