# Peer review of "The Oral Health of a Group of 19th Century South Australian Settlers in Relation to Their General Health and Compared with That of Contemporaneous Samples"

_dentistry, 2023, doi:10.3390/dj11040099_

Round 1

Reviewer 1 Report

Table 1. Can you please add the meaning of letters M, F, U under the table like in Table 3.

Why is SMB 72 highlighted?

Author Response

Authors' Response: Thank you for your suggestion. The meanings for the letters ‘M, F, and U’ have been added below Table S1 of the supporting material. Unfortunately, SMB 73 was highlighted in error and this has now been rectified.

Reviewer 2 Report

The article is very clear and interesting. The tables and images are high quality and easy to read/understand. All the sections provide enought inoformation about the topic. Limitations are well explained and coherent with the research difficulties, particularly i appreciated the one about enmal hypomineralization. Overall the article could provide an overview of south australian settlers oral health that could improve the australian historical medical records/archives and set a point of reference for similar future researches.  

Author Response

Authors’ Response: Thank you for your comments. We hope that this will research article will become a foundational paper on the oral health of colonial South Australian settlers. We also hope that the data included in this paper will be useful for other researchers and will become highly cited.

Reviewer 3 Report

REVIEWER'S REPORT

MANUSCRIPT ID dentistry-2237857

First of all, congratulations to the authors for the conduction of this study research. These findings could serve as solid basis for the initial understanding and further studying of oral health of specific human populations across the world from the near past, including the proposition of the non-invasive methodology as well.

There are few things to improve, for the manuscript to be more clearly written.

1.     Subsection 2.1 Materials – The archaeological sample (lines 78-95) – the suggestion for the authors should be to try to simplify its explanation in fewer sentences.

2.     Although the term «subadult» is perhaps appropriate one for the archaeological samples, it would be more appropriate to explain and relate it to more specific age groups of found remnant sceletons. In the epidemiology of oral diseases it is more appropriate to relate the diseases to more specific age periods, also considering the time when they have lived.

3.     Authors should be careful in naming and ordering the tables in the manuscript. There is Table 4 that has appeared for two times in the text, and there is no Table 6 inserted anywhere, although it was mentioned in the text for several times.

4.     Also, authors should have been encouradged to try to reduce the number of tables in the manuscript and in supplement files, perhaps by relating those with similar content. It is pretty hard to follow all of presented table content.

Author Response

Reviewer 3: Point 1. Subsection 2.1 Materials – The archaeological sample (lines 78-95) – the suggestion for the authors should be to try to simplify its explanation in fewer sentences.

Authors’ Response: Point 1. Thank you for the suggestion. The text for subsection 2.1. Materials have been revised and reduced to clarify the description of the archaeological sample investigated.

Reviewer 3: Point 2. Although the term «subadult» is perhaps appropriate one for the archaeological samples, it would be more appropriate to explain and relate it to more specific age groups of found remnant skeletons. In the epidemiology of oral diseases it is more appropriate to relate the diseases to more specific age periods, also considering the time when they have lived.

Authors’ Response: Point 2. Revisions of the text for subsection 2.1. Materials continued. Clarification of the terminology used to identify the different age ranges/ groups of the individuals within this sample has been added to the text, with examples for the readers. The ages are now specified. The word ‘subadult’ has been used for this study as the sample is from an archaeological context and this research will appeal to a broad multidisciplinary audience.

Reviewer 3: Point 3. Authors should be careful in naming and ordering the tables in the manuscript. There is Table 4 that has appeared for two times in the text, and there is no Table 6 inserted anywhere, although it was mentioned in the text for several times.

Authors’ Response: Point 3. Thank you for pointing this out. The text in subsection 3.8. Comorbidities, and the Table following this text (now Table 6), have been revised and the errors have been corrected. The entire manuscript has also been checked to confirm that Table 6 is now correctly cited.

Reviewer 3: Point 4. Also, authors should have been encouradged to try to reduce the number of tables in the manuscript and in supplement files, perhaps by relating those with similar content. It is pretty hard to follow all of presented table content.

Authors’ Response: Point 4. The reviewer has kindly commented that this paper willserve as a solid basis for the initial understanding and further studying of oral health of specific human populations across the world from the near past, including the proposition of the non-invasive methodology as well”. The authors feel that the inclusion of data as tables within this paper and the supplementary material would be integral to the understanding of the oral health of this sample and would provide a reference point for future researchers. The authors anticipate that this research article will become a highly cited paper. We note that Reviewers 1 and 2 were content with the inclusion of all the tables and supplementary material. Some colleagues may wish to use detailed findings for comparison with other samples they investigate.